# Robin3D🐦: Improving 3D Large Language Model via Robust Instruction Tuning

## Abstract

Recent advancements in 3D Large Language Models (3DLLMs) have highlighted their potential in building general-purpose agents in the 3D real world, yet challenges remain due to the lack of high-quality robust instruction-following data, leading to limited discriminative power and generalization of 3DLLMs. In this paper, we introduce Robin3D, a powerful 3DLLM trained on large-scale instruction-following data generated by our novel data engine, Robust Instruction Generation (RIG) engine. RIG generates two key instruction data: 1) the Adversarial Instruction-following data, which features mixed negative and positive samples to enhance the model's discriminative understanding. 2) the Diverse Instruction-following data, which contains various instruction styles to enhance model's generalization. As a result, we construct **1 million** instruction-following data, consisting of 344K Adversarial samples, 508K Diverse samples, and 165K benchmark training set samples. To better handle these complex instructions, Robin3D first incorporates Relation-Augmented Projector to enhance spatial understanding, and then strengthens the object referring and grounding ability through ID-Feature Bonding. Robin3D consistently outperforms previous methods across five widely-used 3D multimodal learning benchmarks, without the need for task-specific fine-tuning. Notably, we achieve a **7.8%** improvement in the grounding task (Multi3DRefer) and a **6.9%** improvement in the captioning task (Scan2Cap).

## 1 Introduction

Spatial Intelligence [Li, 2024] refers to the ability of AI to understand the 3D world and reason within 3D space. Related ideas, such as Embodied AI [Duan et al., 2022] and Robotic Agent [Bousmalis et al., 2023], express a similar aim to build general-purpose assistants in the 3D real world. To achieve this goal, researchers have drawn inspiration from the success of 2D Multimodal Large Language Models (MLLMs) [Liu et al., 2024c; You et al., 2023] and have started exploring the potential of 3D Large Language Models (3DLLMs) to create general agents [Hong et al., 2023; Chen et al., 2024a; Wang et al., 2023b; Huang et al., 2023a] in the 3D domain or to attain Spatial Intelligence.

Instruction-following tuning [Liu et al., 2024c;a;b] in MLLMs refers to training the LLM to execute natural language commands by integrating both textual and visual information. In contrast to the versatile image-text pairs employed for training 2D MLLMs, collecting 3D instruction-following data for 3DLLM remains a significant challenge. Although existing works have made progress [Hong et al., 2023; Huang et al., 2023b; Chen et al., 2024c] in generating more instruction data, they still lack robustness in two aspects: 1) Most of the existing instruction data consist of positive pairs, lacking adversarial or negative samples. Therefore, models trained on such data tend to be less discriminative because they might overfit to the positive pairs and are more likely to hallucinate positive responses to any input. 2) Some instruction data also lack diversity in language styles, as human annotators or generative models [OpenAI; Wang et al., 2023a] are typically asked to follow fixed instructions when describing objects [Chen et al., 2020; 2021], or the data is generated using predefined templates [Achlioptas et al., 2020], which may limit models' generalizability.

To address these challenges, we introduce Robin3D, a robust and powerful 3D Large Language Model tuned on large scale instruction-following data generated by our novel data engine, Robust Instruction Generation (RIG) engine. Specifically, RIG is designed to generate two types of data:

❶ Adversarial Instruction-following data, which is characterized by mixing adversarial or negative samples with positive ones. This process decouples the potential memorized positive pairs in the training set, leading to a more discriminative understanding of individual objects and instructions. To present a comprehensive adversarial dataset, we cover both object-level and scene-level instructions, from category-based identification problems to expression-based reasoning challenges, resulting in four new tasks. ❷ Diverse Instruction-following data, which first comprehensively collects various types of instructions from existing studies or transforms current tasks into instruction-following format. To harness the powerful in-context learning capability of LLMs, we use ChatGPT [OpenAI] to diversify the language styles of the instructions by crafting specific prompt engineering templates tailored to each task. Combining these with the original training sets of current benchmarks, we construct **1 million** instruction-following samples, with approximately 344K adversarial data (34%), 508K diverse data (50%), and 165K benchmark data (16%), as shown in Fig. 1 (right).

Akin to current SOTA approaches [Huang et al., 2023a; Huang et al.], our Robin3D model uses pre-trained 3D [Zhou et al., 2023] and 2D [Oquab et al., 2023] models to effectively represent 3D scenes at the object level, leveraging object IDs [Huang et al., 2023a] to refer to the objects in the input or ground them in the output. However, the original methods inevitably weaken the spatial relationships between object features due to the object-centric normalization in

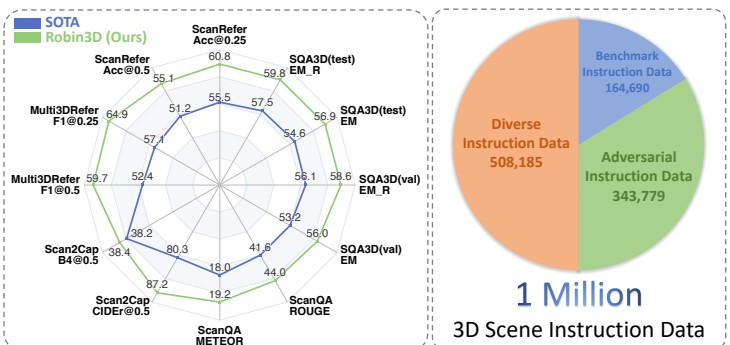

Figure 1: Robin3D surpasses previous SOTA across all the benchmarks (*left*), by training on our RIG-generated 1 million data (*right*).

their 3D backbone [Zhou et al., 2023], which hinders the learning of our diverse visual grounding data. Moreover, in the current approaches [Huang et al., 2023a; Huang et al.], object IDs are simply appended to object features in an interleaved manner, which may lack sufficient connection between the IDs and the features, making it struggle with complex referring and grounding requirements in our adversarial instruction data. To tackle these problems, Robin3D first integrates the Relation-Augmented Projector (RAP) to enhance spatial relationship understanding by enriching object-centric features with scene-level context and positional information. We then introduce ID-Feature Bonding (IFB), which bonds each ID to its corresponding features by wrapping the features with identical ID tokens. IFB further reinforces this connection by a post-vision order, resulting in a more informative ID-feature sequence to improve the referring and grounding ability.

We conduct comprehensive experiments across five representative 3D multimodal learning benchmarks, including ScanRefer [Chen et al., 2020], Multi3DRefer [Zhang et al., 2023], Scan2Cap [Chen et al., 2021], ScanQA [Azuma et al., 2022], and SQA3D [Ma et al., 2022]. As shown in Fig. 1 (left), our Robin3D achieves state-of-the-art performance across all the benchmarks without fine-tuning on specific tasks, making a significant step towards Spatial Intelligence.

In summary, our contributions are threefold: (*i*) We introduce Robin3D, a powerful 3DLLM trained on robust instruction-following data generated by our novel data engine, RIG. (*ii*) We incorporate two modules into Robin3D, RAP and IFB, to enhance its spatial understanding as well as its referring and grounding capabilities when handling our complex instructions. (*iii*) Our Robin3D achieves state-of-the-art performance across five widely-used benchmarks without task-specific fine-tuning.

## 2 RELATED WORK

**3D Vision-Language Learning** Recent advancements in 3D vision-language (3D-VL) learning [Chen et al., 2020; Achlioptas et al., 2020; Azuma et al., 2022; Chen et al., 2021; Kang et al., 2024b] have focused on bridging the gap between 3D scene understanding and natural language,

which is essential for developing embodied intelligence. Similar to 2D vision-language learning [Kazemzadeh et al., 2014; Kang et al., 2024c; Johnson et al., 2016; You et al., 2023; Kang et al., 2024a; Antol et al., 2015; Zhang et al., 2024; Kang et al., 2024d], a variety of tasks such as 3D Visual Grounding [Chen et al., 2020; Zhang et al., 2023; Achlioptas et al., 2020], 3D Dense Captioning [Chen et al., 2021], and 3D Question Answering [Azuma et al., 2022; Ma et al., 2022] have been proposed to evaluate the ability of human instruction following in relation to 3D object properties and spatial relationships. Initial efforts focus on building a task-specific model for a single tasks, such as EDA [Wu et al., 2023] for grounding and Vote2Cap-DETR++ [Chen et al., 2024b] for captioning. Recent research has shifted towards developing unified models capable of handling multiple 3D scene-language tasks. Approaches like 3DJCG [Cai et al., 2022] and D3Net [Takahashi & Mitsufuji, 2020] leverage task synergies, while 3D-VisTA [Zhu et al., 2023], 3D-VLP [Jin et al., 2023] and PQ3D [Zhu et al., 2024] introduce pre-training techniques and unified representations to align 3D visual data with language. However, their dependence on task-specific heads restricts their flexibility for more generalized user-assistant interactions.

**3D Large Language Model** Following the success of 2D Multimodal Large Language Models [You et al., 2023; Zhang et al., 2024; Liu et al., 2024c;a;b], researchers begin to explore the reasoning abilities and promising generalizability of LLMs in human instruction following within the 3D domain, leading to the emergence of 3D Large Language Models (3DLLMs). Models like PointLLM [Xu et al., 2023] and Imagebind-LLM [Han et al., 2023] show strong performance in object-level tasks by mapping 3D data into LLMs. However, they face difficulties in handling scene-level reasoning. 3D-LLM [Hong et al., 2023] incorporates positional embeddings and location tokens, and Oryx [Liu et al., 2024d] offers a solution to support multi-view arbitrary resolution image. However, their reliance on 2D encoders limits its ability to fully capture 3D spatial structures. Models such as LL3DA [Chen et al., 2024a], Chat-3D [Wang et al., 2023b], LEO [Huang et al., 2023b], and Scene-LLM [Fu et al., 2024] have made progress in improving scene-level dialogue capabilities, showing promising results in question-answering and captioning tasks. However, their insufficient visual grounding capability limits their application in Embodied AI or Robotic Agents, which require precise object localization and manipulation following human instruction. To further enhance grounding abilities, Grounded 3D-LLM [Chen et al., 2024c] introduces referent tokens and contrastive learning to unify grounding and textual responses. Similarly, Chat-3D v2 [Huang et al., 2023a] proposes the use of object identifiers (object IDs) for referring and grounding. Building on Chat-3D v2[Huang et al., 2023a], Chat-Scene [Huang et al.] further incorporates DINO v2 [Oquab et al., 2023] to provide strong multi-view, object-centric 2D representations. Despite these advancements, current 3D LLMs, which are trained solely on positive 3D visual-language pairs and template-based instructions, suffer from suboptimal generalization and a potential for overfitting.

## 3 METHODOLOGY

### 3.1 PRELIMINARY

To train a 3D LLM using instruction fine-tuning, we first represent the 3D scene as a sequence of vision tokens, then append it with system and question prompts, expressed as sequences of language tokens, to indicate the task. Taking the above tokens as input, a LLM is supervised to output the answer tokens via next token prediction. Specifically, as shown in Fig. 2, given the point cloud of a 3D scene, we use the pre-trained 3D segmenter Mask3D [Schult et al., 2023] to extract object features along with their corresponding 3D masks. Following Huang et al., we further sample each object's point cloud based on the 3D masks, normalize it, and employ the pre-trained Uni3D [Zhou et al., 2023] to extract unified object-centric 3D features. Additionally, 2D masks projected from the 3D masks are used to sample and average 2D features, which are extracted by DINO v2 from multi-view images of each object. Our Relation-Augmented Projector fuses the 3D features and position embeddings from Mask3D and Uni3D into our final 3D features. In line with Huang et al., we incorporate special tokens $\{< \text{OBJ}_i >\}_{i=1...n}$ as object IDs into the vocabulary. These ID tokens are paired with 2D & 3D object features to indicate each object, for referring to the object in the input instruction or grounding the object in model's output. Our ID-Feature Bonding combines each object feature with its corresponding object ID, and appends the system and question prompts at the beginning of the sequence, which are then fed into the LLM. For more details on the object IDs and the extraction of different types of features, please refer to Huang et al..

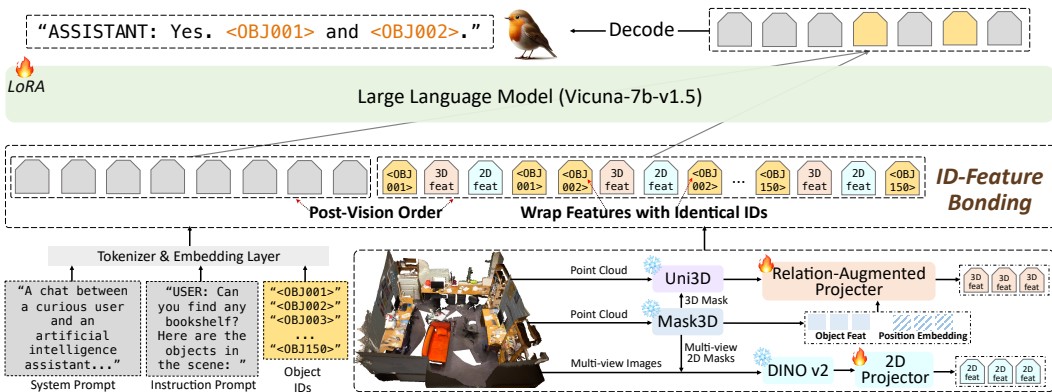

Figure 2: Overview of the proposed Robin3D model structure. *Bottom*: Our Relation-Augmented Projecter fuses the features and position embedding from Mask3D and Uni3D to generate final 3D features. 2D features from DINO v2 are projected into the LLM space. We freeze the Mask3D, Uni3D, and DINO v2. *Middle*: ID-Feature Bonding enhances the connection between object IDs and object features by wrapping the features with identical IDs and the Post-Vision order. *Top*: We use LoRA to fine-tune the LLM on our constructed 1 million instruction data.

## 3.2 RELATION-AUGMENTED PROJECTER (RAP)

As shown in Fig. 2 (bottom), to obtain relation-aware 3D features while preserving the unified object-centric characteristics, our RAP considers three types of 3D features: a) the object features from Mask3D, $X_{mask3d}$, which are scene-level respresentations, containing spatial relationship, as they come across multiple cross-attention layers to exchange information. b) the position embeddings of the Mask3D, $X_{pos}$, which are directly projected from the coordinates of the corresponding object features. c) the unified object features, $X_{uni3d}$, from Uni3D. Our RAP is formulated as:

$$X = \text{Concat}(\text{Norm}_{L2}(X_{uni3d}), \text{Norm}_{L2}(X_{mask3d})), \quad X_{rap} = \text{MLP}(X) + \text{MLP}(X_{pos}) \quad (1)$$

where $\text{Norm}_{L2}$ is the L2 normalization, Concat is the concatenation alongside the channel dimension, and MLP is a multi-layer perceptron with GELU activation [Hendrycks & Gimpel, 2016]. The $X_{rap}$ represents our final relation-aware unified 3D features, which is augmented by the spatial relationship information from Mask3D and the position embeddings.

## 3.3 ID-FEATURE BONDING (IFB)

We propose IFB for better referring and grounding in our instruction-following data by improving the connection between object IDs and object features. As shown in Fig. 2 (middle), we first use two identical ID tokens to wrap the object features. Adhering to the causal attention nature of the LLM, this approach links ID information to the object features via the first ID token, and links object information to its ID via the second ID token. Secondly, we propose a post-vision order, which places the vision tokens at the end of the input sequence, closer to the answer tokens generated by the model. This approach mitigates attention deviation from the answer tokens to the ID-Feature pairs, a problem caused by their relative token distance and rotary position embeddings [Su et al., 2024; Ma et al., 2023], while reinforcing the visual information for improved answer generation. The post-vision order is structured as: [*<System tokens>, <Instruction tokens>, <Vision tokens>, <Answer tokens>*], where <Vision tokens> comprises the ID tokens and object feature tokens.

## 4 ROBUST INSTRUCTION GENERATION (RIG)

### 4.1 ADVERSARIAL DATA GENERATION

The Adversarial data is designed to challenge the model's discriminative capabilities by introducing adversarial or negative samples, ranging from the object-level to the scene-level. It features both category-based identification tasks and expression-based reasoning challenges. As shown in Fig. 3,

we ultimately formulate four novel tasks: Hybrid Object Probing Evaluation (Sec. 4.1.1), Hybrid Referring Object Classification (Sec. 4.1.2), Partial Factual 3D Visual Grounding (Sec. 4.1.3), and Faithful 3D Question Answering (Sec. 4.1.4). Details of each type are as follows:

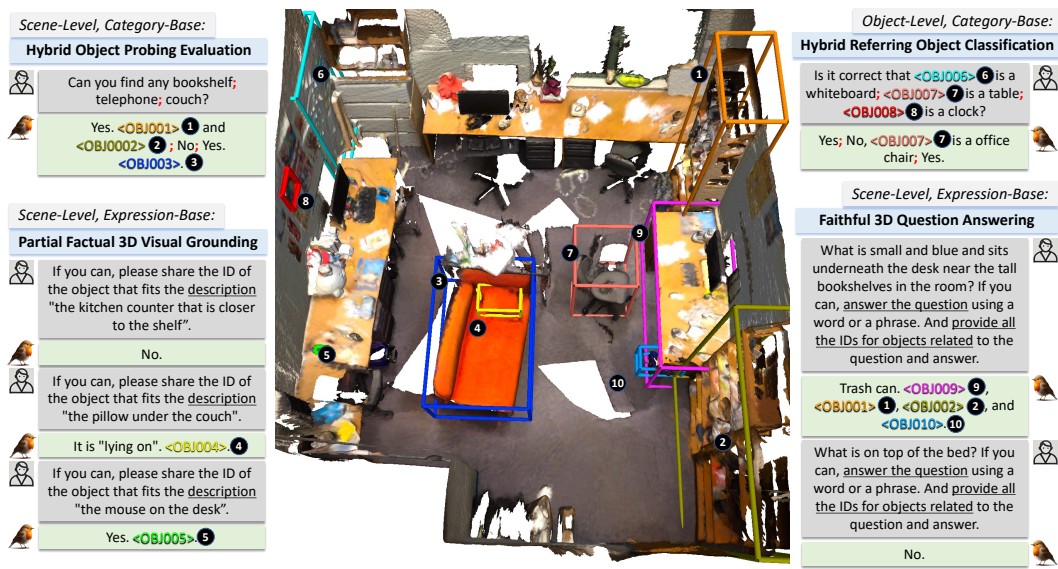

Figure 3: The visualization of examples of adversarial / negative data. For better visualization, we associate each object ID with the same color as its bounding box. The black solid circles with numbers are solely for visualization purposes and are not included in the actual data.

### 4.1.1 HYBRID OBJECT PROBING EVALUATION (HOPE) – *Fig. 3(upper left)*

To construct a scene-level category-based task, we introduce HOPE, which is inspired by the POPE benchmark [Li et al., 2023] in 2D domain. POPE evaluates the tendency of 2D MLLMs to hallucinate by asking yes/no questions about the presence of one specific object at a time. Building on this, HOPE further incorporates such hallucination challenges into the training stage in the 3D domain, aiming to train our model to be more discriminative. Additionally, HOPE presents a hybrid scenario, introducing greater complexity to further advance the decoupling of memorized positive vision and language pairs. Specifically, given a 3D scene, we ask the model to determine the presence of various randomly specified objects. The objects may or may not be present in the scene, and each existing object might have one or more instances. The model is required to answer "*No*" when the object is not present in the scene, and answer "*Yes*" with the object ID of each instance of the object when it exists. As shown in Fig. 3 (upper left), the question combines multiple objects, separated by semicolons ("*;*"), and the answer combines responses for each object, also separated by semicolons. This structure creates a challenging setting that involves hybrid recognition of both positive and negative object presence, combined with multi-instance object localization.

### 4.1.2 HYBRID REFERRING OBJECT CLASSIFICATION (HROC) – *Fig. 3(upper right)*

Referring Object Classification [You et al., 2023] evaluates a model's ability to understand a referred region in 2D domain, focusing on a classification problem by "Region-in Text-out" format. Our HROC dataset extends this task into the training data for 3D domain to create an object-level category-based task, by incorporating adversarial and hybrid challenges. Specifically, in a 3D scene, we randomly create hybrid positive and negative ID-Category pairs to form our questions, as illustrated in Fig. 3 (upper right). A positive pair consists of a valid object ID and the ground truth category. The bounding box of the corresponding object ID must overlap with one ground truth bounding box, and the category of the overlapping object is defined as the ground truth category. A negative pair includes a valid object ID and a randomly selected category that is present in the scene but not the ground truth category to serve as an adversarial challenge. The model is required

to answer "*Yes*" for positive pairs and "*No*" with the correct category for negative pairs. The pairs and corresponding answers are separated by semicolons (";").

### 4.1.3 PARTIAL FACTUAL 3D VISUAL GROUNDING (PF-3DVG) – *Fig. 3(lower left)*

Our PF-3DVG introduces a scene-level expression-based task, featuring three types of data in 3DVG: unfactual data, partially factual data, and factual data. For unfactual data, given a 3D scene, we randomly select a reference from Sr3D+ [Achlioptas et al., 2020] where the indicated object does not exist in the scene. The model is required to answer "*No*" when prompted with the question, as shown in the first example of Fig. 3 (lower left). For partial factual data, given a reference from Sr3D+ and its corresponding 3D scene, we randomly switch the described spatial relationship with a different one based on the predefined template of Sr3D+. For example, as shown in the second example of Fig. 3 (lower left), we change the original reference "*the pillow lying on the couch*" to "*the pillow under the couch*". In this case, it is still possible for human annotators to ground the target based on this partial factual information, but this introduces an adversarial challenge since the information is not completely accurate. Therefore, we require the model to retify the information and answer "*It is 'lying on'*" while providing the grounding result (object ID). Notably, we only use references whose target object has no distractors sharing the same category, ensuring that the partial factual information is still informative enough for grounding the target and does not lead to ambiguity. For factual data, we randomly augment the spatial relationship with its synonym to improve diversity. For example, the synonym of "*below*" can be "*under*", "*beneath*", or "*underneath*".

### 4.1.4 FAITHFUL 3D QUESTION ANSWERING (3DFQA) – *Fig. 3(lower right)*

The original 3D Question Answering (QA) task [Azuma et al., 2022] includes only positive samples, which can potentially lead to the model memorizing fixed combinations of 3D scenes and QA pairs. To address this, we propose Faithful 3D Question Answering, a scene-level expression-based task which incorporates both negative and positive examples with an additional grounding requirement. To construct negative samples, we first sample a QA pair and collect the related objects that are mentioned in the question or the target objects of the answer from Azuma et al. [2022]. Then, we randomly select a 3D scene that lacks those related objects. A new instruction is added to the question: "*If you can, answer the question... and provide all the IDs...*" as illustrated in Fig. 3 (lower right). In this case, the model must faithfully answer "*No*" based on the absence of related objects in the 3D scene and must not provide any object IDs, demonstrating its reliance on the scene for making decisions. For positive samples, directly taken from Azuma et al. [2022], the model must answer the question while faithfully grounding its "evidence" for the answer, i.e., providing the IDs of the related objects. Therefore, the model trained on our 3DFQA dataset is forced to generalize beyond memorization, learning to respond faithfully to both positive and negative samples.

## 4.2 DIVERSE DATA GENERATION

The Diverse data aim to enhances the model's generalization by first incorporating multiple different types of instruction-following data and then increasing the linguistic diversity of the instructions.

We first collect large scale data from different tasks outside the benchmark dataset. Specifically, given a 3D scene, we collect question-answering pairs from the following tasks: 1) Category Question-Answering task from Huang et al. [2023a], where the model is asked to answer the category of a specified object. 2) Nr3D Captioning task from Huang et al. [2023a], where the model is asked to caption the spatial relationship of a specified object to its neighbor. The ground truth is constructed from Nr3D [Achlioptas et al., 2020] dataset. 3) Appearance Captioning task from Chen et al. [2024c], where the model is asked to captioning the physical attributes or visual characteristics of a specified object. The ground truth is generated by CogVLM [Wang et al., 2023a]. 4) Region Captioning task from Chen et al. [2024c], where the model is asked to caption the region encircling a specified object. The ground truth is organized by ChatGPT [OpenAI]. 5) End to end 3D Visual Grounding from Nr3D dataset [Achlioptas et al., 2020], where the model is not provided ground truth candidates, different from the original setting in Nr3D. 6) End to end 3D Visual Grounding from Sr3D+ dataset [Achlioptas et al., 2020], where the model is also not provided ground truth candidates, different from the original setting in Sr3D+.

| Assistant => | $D_{scanrefer}$ |

| $\mathcal{D}_{task}$ | In context examples | |
|---|---|---|
| ScanRefer | User => *The desk, it is placed close to the wall, in front is a black board, to the right is another table. Behind is a chair.* | Assistant => *The desk is placed close to the wall, with a blackboard in front, another table to the right, and a chair behind.* |
| Multi3dref | User => *The sink is located on the vanity to the right of the vacuum cleaners.* | Assistant => *To the right of the vacuum cleaners, the sink is set on the vanity.* |
| Nr3D | User => *The plant at the far right hand side of the bookcase tucked in the furthest corner of the desk.* | Assistant => *At the extreme right end of the bookcase, the plant is nestled in the deepest corner of the desk.* |
| Sr3D+ | User => *Find the office chair that is near the copier.* | Assistant => *Locate the office chair positioned close to the copier.* |
| ScanQA | User => *What color table is on the left side of the cabinet?* | Assistant => *What is the color of the table positioned on the left-hand side of the cabinet?* |
| SQA3D | User => *I stand looking out of the window and a radiator is right in front of me. What color is the desk to my right?* | Assistant => *With a radiator directly in front of me as I look out the window, what color is the desk to my right?* |

Figure 4: (Upper) Pipeline to generate our Diverse Instruction data by the in-context learning of ChatGPT. (Lower) The one-shot examples for ChatGPT to rephrase the instruction-following data.

To diversify the wording style, we develop a scalable pipeline by harnessing ChatGPT's [OpenAI] in-context learning ability to rephrase the above data. This is achieved through a combination of one-shot examples and structured prompt engineering, as shown in Fig. 4(upper). Formally speaking, given a collected instruction-following dataset $\mathcal{D}_{task}$, where $task \in$ {ScanRefer, Multi3DRefer, Nr3D, Sr3D+, Nr3D Captioning, ScanQA, SQA3D, PF-3DVG, 3DFQA}, we construct a system prompt, $\mathcal{P}_{system}$, to indicate the rephrase requirement and structured output format to ChatGPT, a one-shot example prompt, $\mathcal{P}_{eg}$, to show a rephrased example and output format for ChatGPT to better understand the requirement, and randomly assign a temperature $\mathcal{T}$ from $[1.1, 1.2, 1.3]$ for ChatGPT to increase the randomness of the output diversity. Our rephrased output, $\mathcal{D}_{rephrase}$, is generated by $\mathcal{D}_{rephrase} = \mathcal{M}(\mathcal{P}_{system}, \mathcal{P}_{eg}, \mathcal{D}_{task}, \mathcal{T})$, where $\mathcal{M}$ is the GPT-4o version of ChatGPT. We provide the details of $\mathcal{P}_{system}$ and $\mathcal{P}_{eg}$ in the Fig. 4(upper) for data of ScanRefer as an example. With our "*sentence=*" and "*rephrase=*" structured prompt, GPT-4o can easily follow the requirement and we can conveniently collect the output by detecing the "*rephrase=*" keywords. In Fig. 4(lower), we provide details regarding the one-shot example for each task. Since Nr3D Captioning is constructed from Nr3D, PF-3DVG is from Sr3D+, and 3DFQA is from ScanQA, we do not provide additional examples for them.

### 4.3 DATA SUMMARY

In summary, our Robust Instruction Generation engine produces two types of data: 1) Adversarial Instruction data: a total of 344K samples, consisting of mixed positive and negative pairs, formulated into four new tasks. 2) Diverse Instruction data: a total of 508K samples, covering multiple tasks and various language styles, diversified through in-context learning from ChatGPT.

Table 1: **Quantitative comparison.** "Task-Specific Training" denotes models trained on a specific task, while "Joint Training" denotes models trained jointly on multiple tasks. Entries in gray denote using ground truth question-relative objects annotations. The best and second best results in a fair comparison are highlighted in **bold** and underline, respectively.

| Model | ScanRefer | | Multi3DRefer | | Scan2Cap | | ScanQA(val) | | SQA3D(val) | | SQA3D(test) | |
|---|---|---|---|---|---|---|---|---|---|---|---|---|
| | Acc@0.25 | Acc@0.5 | F1@0.25 | F1@0.5 | B-4@0.5 | C@0.5 | M | R | EM | EM-R | EM | EM-R |
| *Task-Specific Training* | | | | | | | | | | | | |
| ScanRefer | 37.3 | 24.3 | - | - | - | - | - | - | - | - | - | - |
| EDA | 53.8 | 41.7 | - | - | - | - | - | - | - | - | - | - |
| Concretenet | 50.6 | 46.5 | - | - | - | - | - | - | - | - | - | - |
| M3DRef-CLIP | 51.9 | 44.7 | 42.8 | 38.4 | - | - | - | - | - | - | - | - |
| Scan2Cap | - | - | - | - | 23.3 | 39.1 | - | - | - | - | - | - |
| Vote2Cap-DETR++ | - | - | - | - | 37.1 | 67.6 | - | - | - | - | - | - |
| ScanQA | - | - | - | - | - | - | 13.1 | 33.3 | - | - | - | - |
| SQA3D | - | - | - | - | - | - | - | - | - | - | 46.6 | - |
| *Joint Training* | | | | | | | | | | | | |
| D3Net | - | 37.9 | - | 32.2 | 35.7 | 62.6 | - | - | - | - | - | - |
| 3DJCG | 49.6 | 37.3 | - | 26.6 | 31.0 | 49.5 | - | - | - | - | - | - |
| 3D-VLP | 51.4 | 39.5 | - | - | 32.3 | 54.9 | - | - | - | - | - | - |
| 3D-VisTA | 50.6 | 45.8 | - | - | 34.0 | 66.9 | 13.9 | 35.7 | - | - | 48.5 | - |
| PQ3D | - | 51.2 | - | 50.1 | 36.0 | 80.3 | - | - | - | - | 47.1 | - |
| 3DLLM | 30.3 | - | - | - | - | - | 14.5 | 35.7 | - | - | - | - |
| Oryx | - | - | - | - | - | - | 15.0 | 37.3 | - | - | - | - |
| LL3DA | - | - | - | - | 36.8 | 65.2 | 15.9 | 37.3 | - | - | - | - |
| LEO | - | - | - | - | 38.2 | 72.4 | 20.0 | 49.2 | - | - | 50.0 | 52.4 |
| Scene-LLM | - | - | - | - | - | - | 16.6 | 40.0 | - | - | 54.2 | - |
| Chat-3D v2 | 35.9 | 30.4 | - | - | 15.5 | 28.1 | 16.1 | 40.1 | - | - | - | - |
| Grounded-3DLLM | 47.9 | 44.1 | 45.2 | 40.6 | 35.5 | 70.6 | 15.2 | 37.1 | - | - | - | - |
| Chat-Scene | 55.5 | 50.2 | 57.1 | 52.4 | 36.3 | 77.1 | 18.0 | 41.6 | 53.2 | 56.1 | 54.6 | 57.5 |
| Robin3D (Ours) | **60.8** | **55.1** | **64.9** | **59.7** | **38.4** | **87.2** | **19.2** | **44.0** | **56.0** | **58.6** | **56.9** | **59.8** |

## 5 EXPERIMENTS

### 5.1 BENCHMARKS AND METRICS

We provide quantitative results on five widely-used 3D multimodal learning benchmarks: ScanRefer [Chen et al., 2020] for 3D Visual Grounding, Multi3DRefer [Zhang et al., 2023] for General 3D Visual Grounding including zero, single and multiple target objects, Scan2Cap [Chen et al., 2021] for 3D Dense Captioning, ScanQA [Azuma et al., 2022] for 3D Question Answering, and SQA3D [Ma et al., 2022] for 3D Situated Question Answering. The vision data are all based on the ScanNet dataset [Dai et al., 2017], which contains real world 3D point clouds across 1,513 indoor scenes with detailed object annotations. All these benchmarks follow the same data split as ScanNet.

We follow the standard evaluation metrics widely adopted in the respective benchmarks. For Scan-Refer, we calculate accuracy at Intersection over Union (IoU) thresholds of 0.25 and 0.5 (Acc@0.25, Acc@0.5). For Multi3DRefer, we use the F1 score with IoU thresholds of 0.25 and 0.5 to measure performance. In Scan2Cap, we apply the CIDEr@0.5 and BLEU-4@0.5 (C@0.5, B-4@0.5) metrics, combining standard captioning metrics with the IoU metric. For ScanQA, the METEOR and ROUGE metrics, denoted as M and R, are employed. Lastly, SQA3D is assessed with exact match accuracy (EM) and its extended form, EM-R, as suggested by LEO [Huang et al., 2023b].

### 5.2 IMPLEMENTATION DETAILS

We extract 150 object features from each 3D scene, along with the corresponding position embeddings and 3D masks generated by Mask3D. The 2D Projector, as shown in Fig. 2, is a two-layer MLP. We use the Vicuna-7B-v1.5 model [Chiang et al., 2023] as our LLM and fine-tune it using LoRA [Hu et al., 2021] (with a rank of 16) by Cross Entropy loss. The global learning rate is formulated as [batch size × base learning rate × number of GPUs] and is set to 0.00064, with a cosine annealing schedule. For our results in Tab. 1, we first train for 2 epochs on the RIG-generated data, and then train for 2 epochs on the benchmark training sets in the second stage. We train for 1 epoch for each stage to efficiently conduct ablation studies of RIG-generated data. For ablation on RAP and IFB, we train for 1 epoch on the benchmark training sets to avoid potential compound effects.

## 5.3 Quantitative Results

We classify current methods into two categories: Task-Specific Training and Joint Training. Task-Specific Training refers to models only trained for a specific task, while Joint Training means training on multiple tasks jointly. Our Robin3D does not conduct task-specific fine-tuning.

- **Task-Specific Training:** As shown in Table 1, models like EDA and M3DRef-CLIP perform well on their respective tasks due to customized model design for the task. However, they lack the ability to generalize to other tasks. Models like Vote2Cap-DETR++ and SQA3D encounter the similar issue. Therefore, they are not suitable to serve as general-purpose 3D AI agents.
- **Joint Training:** Benefiting from sharing the knowledge across multiple tasks, models like 3D-VisTA and PQ3D show decent performance across multiple tasks, but their dependence on task-specific heads restricts their generalizability. Models like LEO and Chat-Scene show promising results by leveraging LLMs, but their sole training on positive pairs and template-based instructions leads to suboptimal generalization.
- **Our Robin3D:** Due to the robust instruction data generated by RIG, Robin3D significantly outperforms previous models across all the benchmarks. Specifically, Robin3D brings a **6.9%** improvement on Scan2Cap CIDEr@0.5 and a **5.3%** improvement on ScanRefer Acc@0.25. Notably, on the evaluation of Multi3DRefer, which contains zero-target cases that are challenging for models to be discriminative and learn to say "*No*", our Robin3D achieves a **7.8%** improvement in F1@0.25 and a **7.3%** improvement in F1@0.5.

## 5.4 Ablation Study

Table 2: **Ablation study on Robust Instruction Generation.** *Benchmark* denotes training on the original training set of the benchmarks. *Adversarial* denotes adding the Adversarial Instruction data to the training set. *Diverse* denotes adding the Diverse Instruction data to the training set.

| Data | ScanRefer Acc@0.5 | Multi3DRefer F1@0.5 | Scan2Cap C@0.5 | ScanQA(val) M | SQA3D(val) EM |
|---|---|---|---|---|---|
| Benchmark | 45.3 | 50.2 | 73.6 | 17.7 | 48.9 |
| + Adversarial | 49.0 | 55.1 | 82.5 | 18.2 | 50.7 |
| + Diverse | 50.6 | 53.1 | 80.5 | 18.0 | 50.9 |
| + Adversarial & Diverse | 51.8 | 56.9 | 84.1 | 18.4 | 52.6 |

Table 3: **Ablation study on our proposed modules:** Relation-Augmented Projecter (RAP) and ID-Feature Bonding (IFB).

| Model | ScanRefer Acc@0.5 | Multi3DRefer F1@0.5 | Scan2Cap C@0.5 | ScanQA(val) M | SQA3D(val) EM |
|---|---|---|---|---|---|
| Baseline | 41.0 | 45.3 | 69.2 | 17.4 | 48.2 |
| + RAP | 42.5 | 46.3 | 72.0 | 17.6 | 48.6 |
| + RAP & IFB | 45.3 | 50.2 | 73.6 | 17.7 | 48.9 |

We perform ablation studies from the perspective of training data and model structure, respectively. We first evaluate the effectiveness of RIG-generated data by progressively adding the Adversarial Instruction data and the Diverse Instruction data to the training set. We then investigate the contribution of RAP and IFB by comparing models with and without these components.

**Robust Instruction Generation (RIG):** As shown in Tab. 2, by adding the Adversarial Instruction data, we observe a consistent improvement across all benchmarks. Specifically, performance on ScanRefer and Multi3DRefer increases by **3.7%** and **4.9%**, respectively. It is worth noting that the performance on Scan2Cap improves by **8.9%**, even though there is not any object captioning data in the Adversarial Instruction data, which highlights its contribution on enhancing the understanding towards each object by challenging the model with mixed positive and negative samples. Additionally, by adding the Diverse Instruction data, we also provide comprehensive improvements. Specifically, descriptions from the original ScanRefer are annotated by human following a fixed instruction template or expression style, which limits the language diversity. In contrast, the Diverse

Instruction data contains various language styles and task formats, which helps the model generalize better, resulting in a **5.3%** improvement on ScanRefer. Finally, by combining both two types of data, we achieve a further improvement, demonstrating the effectiveness of RIG-generated data.

**Relation-Augmented Projecter (RAP) and ID-Feature Bonding (IFB):** As shown in Tab. 3, the integration of RAP leads to steady improvements across all benchmarks. Notably, the performance on Visual Grounding tasks, including ScanRefer and Multi3DRefer, shows significant gains due to RAP's enhanced spatial comprehension. When IFB is incorporated, further improvements are observed, emphasizing the importance of refining the model's object referring and grounding capabilities by reinforcing the connection between IDs and features.

## 5.5 QUALITATIVE RESULTS

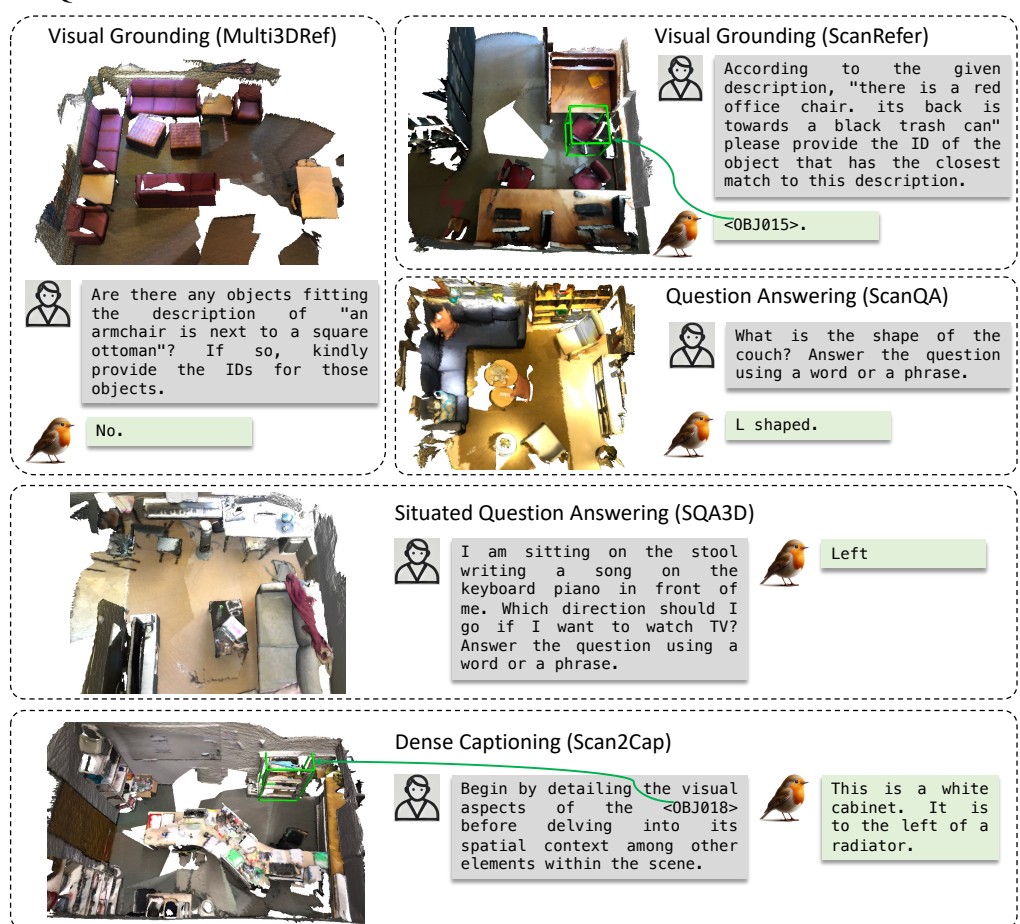

Figure 5: Visualization of Robin3D's responses on all the five benchmarks.

We provide visualization of Robin3D's responses on all the benchmarks in Fig. 5 with the prompts of each task. These results demonstrates the generalization ability of Robin3D on various tasks.

## 6 CONCLUSION

To build a general-purpose AI agent in the 3D real world, we identify the problem of a lack of robust instruction training data in current 3DLLMs. To tackle this challenge, we introduce Robin3D, a powerful 3DLLM trained on large-scale instruction-following data generated by our novel data engine, Robust Instruction Generation (RIG) engine. We generate and collect 1 million instruction data, including benchmark data, adversarial data, and diverse data. To better handle these complex instructions, Robin3D incorporates a Relation-Augmented Projector to enhance the understanding of spatial relationships among objects, and ID-Feature Bonding for better object referring and grounding. Finally, Robin3D achieves state-of-the-art performance across five widely-used 3D multimodal learning benchmarks, making significant progress towards Spatial Intelligence.

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
