# OpenReview forum: "Robin3D: Improving 3D Large Language Model via Robust Instruction Tuning"
_ICLR.cc/2025/Conference — ICLR 2025 Conference Withdrawn Submission_

### Official Review · Reviewer_RG9D · 2024-10-30

**Soundness:** 3
**Presentation:** 3
**Contribution:** 4
**Rating:** 6
**Confidence:** 3

**Summary:**

This paper first constructs 1 million instruction-following data, including adversarial samples and diversified samples to bridge the drawbacks of existing 3D MLLM instruction following fine-tuning datasets. To better handle the proposed complex instructions, this paper first incorporates Mask3D and Relation-Augmented Projector to enhance spatial understanding, and then improve the object referring and grounding ability through ID-Feature Bonding. The trained model Robin3D shows superior performance across five widely used 3D multimodal learning benchmarks.

**Strengths:**

1. This paper constructs a large instruction-following fine-tuning dataset containing adversarial and diverse samples.
2. The zero-shot performance improvement of the trained Robin3D appears evident across various benchmarks and the ablation experiments clearly demonstrate the gains of different designs in the paper.
3. The writing of the article is fluent and easy to understand.

**Weaknesses:**

1. The related work section lacks clarity on the novelty and advantages of the RAP and IFB modules in comparison to existing studies.
(1) Explain how object IDs are linked to object features in previous research and discuss the benefits of wrapping these features with identical ID tokens before and after them.
(2) Describe how earlier studies extract and utilize 3D and 2D features, and highlight the advantages of introducing Mask3D information using RAP.

2. How will the relative proportions of diverse and adversarial samples generated with RIG affect the performance of Robin3D?
Please conduct ablation studies to examine and analyze how datasets with varying proportions of adversarial and diverse samples influence Robin3D's performance across different tasks.


3. If the dataset constructed in this paper is used to fine-tune existing task-specific or joint trained models, will it provide consistent performance gains?
The authors could consider selecting 1 or 2 task-specific and jointly trained models, respectively, and tuning them on the proposed instruction-following tuning dataset to further demonstrate the contribution of this dataset to the community.

**Questions:**

Please refer to the weakness section.

---

### Official Review · Reviewer_jF4G · 2024-11-01

**Soundness:** 3
**Presentation:** 3
**Contribution:** 2
**Rating:** 5
**Confidence:** 3

**Summary:**

This paper introduce a 3DLLM Robin3D trained on their proposed dataset generated by RIG, a pipeline to acquire diverse and discriminative data. Robin3D incorporates two key modules, the Relation-Augmented Projector (RAP) and ID-Feature Bonding (IFB), to enhance spatial understanding and object grounding. The model demonstrates state-of-the-art performance across five 3D multimodal learning benchmarks without task-specific fine-tuning.

**Strengths:**

1. The RIG engine's capability to generate adversarial and diverse instruction data significantly enhances the robustness and generalizability of 3DLLMs. The innovative proposal of adversarial data may help mitigate the hallucination tendencies of large models. The collection of diverse instructions, expanded by GPT to enrich the diversity of expressions, may alleviate the issue of rigid model outputs.
2. The integration of RAP and IFB modules improves the model's spatial understanding and object grounding capabilities.
3. Robin3D achieves superior performance across multiple benchmarks, showcasing its effectiveness and versatility.
4. The models are trained on the whole task data, rather than on individual tasks.

**Weaknesses:**

1. The module's innovativeness is found to be lacking: RAP utilizes linear layers to separately connect the 3D features from the scene, individual object 3D features, and positional information features, followed by concatenation. A possible baseline (chat-scene) employs the exact same encoders, using linear layers to connect 3D features and positional features, and then concatenating individual object 3D features. The only modification made is the interchange of inputs to the linear layers. Similarly, IFB introduces an ending ID token to signal the end of an object presentation, followed by a rearrangement of vision tokens and prompt tokens. This method of simply altering the prompt tokens is not particularly innovative.

2. Are the results state-of-the-art (SOTA): The experimental results compared against the Chat-Scene model have been open-sourced and are being continuously updated. Prior to the submission deadline for this conference, the accuracy on the ScanRefer dataset had already surpassed 61% and 55%, outperforming the method proposed in this paper. This paper should have utilized the most recent projects and results as benchmarks; otherwise, the effectiveness of the proposed method cannot be ascertained.

3. Model generality: Mainstream approaches about 3DLLMs typically employ a single, well-trained model to evaluate performance across multiple tasks. The joint training without task-specific fine-tuning method described in the paper does not represent a contribution unique to this work.

**Questions:**

1. This paper employs the same detectors and 2D&3D encoders as chat-scene (Mask3D, Uni3D, DINO-v2). What are the significant innovations of this model compared to chat-scene?

2. In Table 1, what do the grey sections referring to "ground truth question-relative objects annotations" specifically indicate? Is the explicit positional information P introduced by the dedicated detector Mask3D on the ScanQA test set considered as "ground truth question-relative objects annotations"?

3. Results of Baseline(+ RAP & IFB) in Table3 are the same as the benchmark results in Table 2. In the "ABLATION STUDY" section, it seems there might be a confusion regarding the order of incorporating modules and datasets. Benchmark(+ Adversarial & Diverse) should include RAP&IFB and encompass all datasets. Why are the results of the ablation study (+ Adversarial & Diverse) inconsistent with the results in Table 1?

---

### Official Review · Reviewer_D2SR · 2024-11-01

**Soundness:** 3
**Presentation:** 3
**Contribution:** 3
**Rating:** 6
**Confidence:** 3

**Summary:**

The paper targets the interesting problem of instruction tuning on 3D LLMs. As there lacks sufficient dataset, the paper introduces a new 1M instruction-tuning datset, which contains 344K adversarial samples, 508K diverse samples as well as 165K benchmark training set samples. Based on the dataset, the proposed algorithm, called Robin3D, obtains promising results in the ground task as well as caption task.

**Strengths:**

1. The paper targets the challening problem of 3D LLM for ground task as well as caption task.
2. To address the problem, the paper presents a robust instruction generation engine and 1M instruction-following data has been presented.
3. The paper obtains promising experimental results on five 3D multimoal learning benchmarks.

**Weaknesses:**

1. Will the 1M 3D instruction dataset be release to the public? The main contribution of the paper lies on the datasets, thus whether the dataset will be released to public is important to evaluate the contribution of the paper.
2. The dataset seems to be designed specifially for the 3D indoor environment. How about the generation ability of the dataset and the model used for the outdoor environment, like the 3D street?
3. Is it possible to provide an ablation study on different of training examples? It would be better to know the model performance with different number of training data.
4. The model is based on Vicuna-7B-v1.5 backbone. How about the performance if other LLM models are utilized? Besides, if larger LLM model is utilized, is a larger training dataset can further boost the performance?

**Questions:**

Please address the questions raised in the weakness section.

---

### Official Review · Reviewer_NEzg · 2024-11-02

**Soundness:** 2
**Presentation:** 3
**Contribution:** 2
**Rating:** 5
**Confidence:** 4

**Summary:**

The paper introduces Robin3D, a powerful 3D Large Language Model (3DLLM) trained using large-scale instruction-following data generated by an innovative Robust Instruction Generation (RIG) engine to address the lack of robustness and diversity in current 3DLLMs' training data.

Besides, Robin3D incorporates two important modules: Relation-Augmented Projector (RAP) and ID-Feature Bonding (IFB). RAP enhances the model's understanding of spatial relationships between objects, while IFB strengthens the connection between object IDs and features, improving the model's referring and grounding capabilities, enabling it to better handle complex instructions.

**Strengths:**

1. Reasonable motivation
    - Expands existing ScanNet 3D text annotations through the data engine.
2. Strong experimental results
   - Demonstrates excellent performance.
3. Clear and complete paper writing.

**Weaknesses:**

I have some questions about this paper that need further discussion. Please see them below.

If the authors can address my concerns, I am willing to raise my score.

**Questions:**

1. The motivation in the paper is somewhat mixed. Although it emphasizes pre-training with adversarial samples, it also highlights improvements through the Relation-Augmented Projector (RAP) and ID-Feature Bonding (IFB), which may seem like an attempt to pad contributions.

2. The ablation study shows that RAP and IFB contribute less to 3D QA (with low improvements in Table 3's ScanQA and SQA3D) but significantly help 3D grounding. Can the authors explain why?

3. The paper lacks details on the prompts used for Adversarial Data Generation and the data creation process. Is the input for adversarial samples only the ground truth prompt?

4. The ablation for Adversarial Data is insufficient, making it unclear whether the performance improvement is due to the increase in data volume or specifically from the adversarial samples.

5. The authors should compare methods like VLM-Grounder[1] and Coarse Correspondences[2] using video as a modality.

6. Should the authors consider extending their approach to non-ScanNet scenes?

7. The pre-training work should provide training configurations and training time.

8. Can the proposed RIG be extended to the point level to enhance point-level LLM performance, such as with PointLLM [3] or GPT-4Point [4]? Additionally, could it be generalized to outdoor 3D LLMs like DriveLM [5] or LiDAR-LLM [6]? It would be beneficial for the authors to discuss this in the paper.

[1] A VLM Agent for Zero-Shot 3D Visual Grounding

[2] Coarse Correspondence Elicit 3D Spacetime Understanding in Multimodal Language Model

[3] PointLLM: Empowering Large Language Models to Understand Point Clouds

[4] GPT4Point: A Unified Framework for Point-Language Understanding and Generation

[5] DriveLM: Driving with Graph Visual Question Answering

[6] LiDAR-LLM: Exploring the Potential of Large Language Models for 3D LiDAR Understanding

---

### Official Review · Reviewer_Jaxq · 2024-11-08

**Soundness:** 3
**Presentation:** 3
**Contribution:** 2
**Rating:** 5
**Confidence:** 5

**Summary:**

This paper introduces Robin3D, a 3D large language model trained to follow instructions in 3D environments using the Robust Instruction Generation (RIG) engine, which creates a one-million-sample dataset. RIG generates Adversarial and Diverse instruction data to improve Robin3D’s discriminative power and generalization. Robin3D employs a Relation-Augmented Projector for spatial understanding and IDFeature Bonding for object grounding, achieving notable improvements over previous models, including a 7.8% gain in grounding and 6.9% in captioning without task-specific fine-tuning.
While Robin3D performs impressively across multiple ScanNet benchmarks, as noted in the weaknesses, some concerns remain regarding its network architecture and experiments.

**Strengths:**

• The paper introduces a large-scale 3D scene-instruction dataset that includes diverse instruction types, integrating varied instruction styles, existing benchmark instructions, and challenging adversarial instructions, enhancing the model’s robustness and generalization.\
• It proposes novel architectures that effectively leverage both 2D and 3D object-centric features, enabling richer spatial understanding and stronger object-grounding capabilities in complex 3D environments.

**Weaknesses:**

• Relies on off-the-shelf 3D instance segmentation models trained on ScanNet with closed-set categories. I recommend the authors to consider Segment3D [1] for open-vocab, class-agnostic segmentation.\
• Cropping instance-level point clouds and applying object-level 3D point cloud CLIP (Uni3D) can limit the receptive fields and be computationally heavy. I recommend the authors to try scene-level CLIP (OpenScene [2], RegionPLC[3]) and then cropping the output features.\
• Table 1 reports only traditional NLP metrics (e.g., BLEU, CIDEr, METEOR, Rouge). I recommend the authors include LLM-based evaluation (e.g., GPT or Mistral) for better alignment with human assessment.\
• The experiments are limited to the ScanNet dataset. I recommend that the authors expand to other datasets (e.g., SceneVerse [4]) for broader evaluation.

[1] Huang et al., "Segment3D: Learning Fine-Grained Class-Agnostic 3D Segmentation without Manual Labels", ECCV, 2024.\
[2] Peng et al., "OpenScene: 3D Scene Understanding with Open Vocabularies", CVPR, 2023.\
[3] Yang et al., "RegionPLC: Regional Point-Language Contrastive Learning for Open-World 3D Scene Understanding", CVPR, 2024.\
[4] Jia et al., "SceneVerse: Scaling 3D Vision-Language Learning for Grounded Scene Understanding", ECCV, 2024.

**Questions:**

- Table 1 does not clarify what kinds of LLM each method uses but it is worth doing that since multimodal LLM performance usually depends on LLM performance. \
- What is the baseline in Table 3? I couldn't find out the network architecture of the baseline.

---

### Note · Authors · 2024-11-14

I have read and agree with the venue's withdrawal policy on behalf of myself and my co-authors.